# Values, Health and Well-Being of Young Europeans Not in Employment, Education or Training (NEET)

**DOI:** 10.3390/ijerph20064840

**Published:** 2023-03-09

**Authors:** Marja Hult, Minna Kaarakainen, Deborah De Moortel

**Affiliations:** 1Department of Nursing Science, Faculty of Health Sciences, University of Eastern Finland, 70211 Kuopio, Finland; 2Department of Health and Social Management, Faculty of Social Sciences and Business Studies, University of Eastern Finland, 70211 Kuopio, Finland; 3International Department, Savonia University of Applied Sciences, 70210 Kuopio, Finland; 4Interface Demography, Department of Sociology, Vrije University Brussels, 1050 Brussels, Belgium; 5Research Foundation Flanders, 1000 Brussels, Belgium

**Keywords:** European Social Survey, Human Values Scale, NEET, self-rated health, subjective well-being unemployment, young

## Abstract

Youth unemployment is a problem that undermines young people’s health and well-being and is also a concern for their immediate communities and society. Human values predict health-related behaviour; however, this relation is very little studied and not examined earlier among NEET (not in employment, education or training) young people. This study aimed to explore the association between four higher-order human values (conservation, openness to change, self-enhancement, self-transcendence), self-rated health (SRH) and subjective well-being (SW) among NEET young men and women (n = 3842) across European regions. Pooled European Social Survey data from 2010–2018 were used. First, we run linear regression analysis stratified by European socio-cultural regions and gender. Then, multilevel analyses by gender with interactions were performed. The results show expected variation in value profiles across genders and regions and corresponding differences in SRH and SW. Significant associations between values and SRH and SW were found among both genders and across the regions; however, the results did not entirely confirm the expectations about the “healthiness” of specific values. More likely, prevailing values in societies, such as the social norm to work, might shape these associations. This study contributes to a deeper understanding of the factors affecting NEETs’ health and well-being.

## 1. Introduction

Since the recession of 2008, the transition from education to work has become more complex. Many young adults have found themselves in uncertain, precarious employment relationships or have not found employment at all [1]. The unemployment rate among 15- to 24-year-olds in Europe was an average of 15.1% in October 2022. The highest rate was in Spain (32.3%), and the lowest was in Czechia (5.7%) [2]. The youth unemployment rate is substantially higher than the general unemployment rate (which is on average 6%), ranging from 12.5% in Spain to 2.1% in Czechia in 2022. Youth unemployment is a significant concern for young people not in employment, education or training (i.e., NEET), their families, societies and the economy [3]. Almost 30% of NEETs are long-term unemployed, meaning the length of unemployment is 12 months or more [4]. Therefore, it is suggested that the concept of NEET should not be used for those in precarious jobs, only briefly unemployed or engaged in self-directed activities such as travelling [5]. Several background factors have been identified as predictors of NEET status, such as low educational level and immigrant background; however, one of the most significant factors is health problems. For example, a disability increases the risk of becoming NEET by 40% compared to healthy young people [3].

This paper focuses on the association between values and health and well-being in a sample of European NEETs. By doing this, we add to the existing knowledge about individual and contextual factors affecting the health and well-being of NEETs to support all the means of development that have the potential to promote their health and well-being. Previous research has focused on the one hand, on the health and well-being of NEETs, and on the other hand, on the association between values and well-being. We believe values predict health-related behaviour; life values guide individuals to make choices and shape their lives. Individuals will develop a sense of their values to live a fulfilling and meaningful life.

### 1.1. Health and Well-Being of NEET Young People

In the general population, the relationship between poor health and well-being and unemployment is explained by social selection to unemployment [6,7,8] or the social causation process [9]. In short, social selection means that unhealthy people fail to get jobs (selected out), and the social causation process highlights the detrimental effects of unemployment on health and well-being (see [10]), for example, through diminished resources. The loss of resources (e.g., energy, professional skills) and the consequent lower job-gaining prospects can be explained by the conservation of resources (COR) theory [11], according to which the loss of resources leads to a loss spiral. Loss of resources thus harms well-being when unemployment is prolonged [12]. Several other models have also been tested to explain the relationship between unemployment and poor health. Jahoda’s [13] latent deprivation model was found to best explain depression in a sample of unemployed individuals [14]. The model states that besides apparent consequences, such as salary, unemployment has latent consequences that go beyond earning a living [13]. The latent consequences of work that the unemployed are deprived of are collective purpose, social contact, time structure, enforced activity, and status and identity. The absence of time structure and regular activities have been shown to strongly predict poor health [15]. However, poor financial resources were also added to the model [16] as an essential predictor of poor health [15], and financial deprivation has since become the most substantial factor in explaining poor health during unemployment [17,18,19].

The relationship between poor health and well-being among NEET youth is explained by both social selection and social causation. The risk of becoming a NEET youth can be seen already in early adolescence. Mental health vulnerability (specifically among girls), substance use-related disorders (specifically among boys) and depressive symptoms and suicidal thoughts in youth precede but are an integral part of NEETs’ lives [20,21,22,23]. In later life, these can be fatal for future employment and career prospects because alcohol-related and psychiatric conditions hinder most from finding employment [6]. On the other hand, intense smoking and bad eating habits are common among NEETs [21,24,25]. Furthermore, unemployment at a young age may lead to a passive lifestyle through perceived hopelessness and giving up, which, with a lack of money, may lead to deteriorating self-confidence, stress-related health problems and negative mood [26]. In addition, mood and behavioural disorders may stem from a lack of awareness if the young person does not know what to do in life (a reason for NEET status). These disorders were found to be more severe in those NEETs who were deliberately without a job, studies or training [21]. Unemployment itself might be a source of shame, and unemployment associated with mental health problems can be highly stigmatising and lead to experiences of discrimination [27]. In addition, blame has its effects on well-being. NEETs feel self-blame, which points to them, and system blame pointing to systemic factors as reasons for their unemployment [28]. Furthermore, NEET young people are also more pessimistic than other youth regarding their probability of surviving in life. Pessimism probably stems from diminished mental health resources, high feelings of deprivation, low self-esteem and low social support that hamper coping with everyday life [29]. Interestingly, a higher level of education among unemployed youth may lead to either better [30] or worse [31] well-being. NEETs with higher education may be stressed because they did not expect to end up unemployed and, therefore, feel disappointed and inconsistent in their status [31].

### 1.2. The Association between Values and Health and Well-Being

Life values are gore beliefs which guide individuals’ behaviour to make choices and shape their lives by enhancing self-motivation [32,33]. Individuals will develop a sense of their values to live fulfilling and meaningful lives [34]. There are multiple ways to define values relating to health and well-being. In this study, we use the Human Values Scale, which is rooted in Schwartz’s [34,35] and his team’s [36] value definitions. The original Human Values Scale includes ten basic and four higher-order values Schwartz [34] defines conservation, openness to change, self-enhancement and self-transcendence as higher-order values. Conservation as a higher-order value includes traditions, conformity and security. These values highlight self-restriction, order and avoiding change [36]. Conformity as a value indicates the person’s ability and willingness to restrain actions, inclinations and impulses that might upset, harm or violate others, social expectations or norms. Schwartz et al. [36] divide this into rules and interpersonal sections linked to people with traditional values respecting and committing to their culture, traditions and religion. Security as a value refers to an attempt to achieve safety, harmony and stability on personal and societal levels [34,36]. Among young people, conservation values may relate to environmental issues and conserving traditional practices and customs [37,38].

Openness to change values refers to a person’s readiness for new experiences, ideas and actions. These values include hedonism, self-direction and stimulation. As a basic value, hedonism defines pleasure and sensual gratification for oneself. Self-direction expresses independent thought and action. Stimulation expresses the excitement, novelty and challenge in life [34,36]. Self-enhancement values include power and achievement and are linked to hedonism. These values contrast with self-transcendence values. Achievement describes personal success and competence. Power is defined by control and dominance over people and resources [34,36]. Personal growth, self-expression, development and achieving goals, as well as improving the quality of life, are essential to young generations [37]. Self-transcendence values highlight persons transcending their interests for the sake of others. Self-transcendence as a high-order value includes benevolence and universalism. The value of benevolence lies in the preservation and enhancement of those people who are closest and in frequent contact. This indicates that the person is interested in their well-being and is willing to be loyal. Universalism emphasises understanding, appreciation and protection of all people’s welfare. Universalism can be seen as concern, nature and tolerance [34,36].

The values of conservation and openness to change are opposing, as are self-enhancement and self-transcendence, and these opposing values are not expressed by the same person. Openness to change and self-transcendence are found to be beneficial, and conservation and self-enhancement harmful, for health and well-being [39,40]. Among NEET young people, openness to change might indicate an active social life and more social support, creativity to deal with problems and flexibility with life changes, which lead to better health and well-being. Self-transcendence, in turn, is associated with better health and well-being because of an orientation towards the well-being of other people and being more trusting. Trust is a significant factor in well-being, since it is shown that social trust (thinking that other people are mainly trustworthy) and institutional trust have positive effects on subjective well-being across Europe; however, the country differences are remarkable [41]. Trust is linked to values in another way as well; for example, in Nordic countries, a high level of social trust is fostered with value similarity, i.e., most people share the prevailing values in their country [42]. The negative effect of self-enhancement might stem from the lack of trust when NEETs value power and dominance over other people and perceive their peers as competitors. However, there will always be others who are more successful, and social comparison leads to poor health and well-being. The lack of status, which means lack of money and a job as signs of success in capitalism, explains poor health according to Jahoda [13].

The association of conservation with poor health and well-being is most likely explained by social norms to work because NEETs are not doing what is socially expected from them, i.e., studying or going to work. In countries with strong social norms to work, the well-being of unemployed people tends to be lower compared to their employed counterparts [9,43,44]. The social norm to work has both external and internal impacts; unemployed people sanction themselves for not following their internalised sense of duty (feeling guilty), and they expect external disapproval because they do not live according to society’s norms (feeling shame) [44]. This plausible pathway explains the lower self-esteem and inferiority among unemployed people [45].

### 1.3. Cultural, Societal and Gender Effects on the Health and Well-Being of the Unemployed

For the research purposes, different classifications of European countries have been presented. For example, Buttler [43] considers school-to-work transition characteristics when defining employment-centred (Western Europe), liberal, universalistic (Northern Europe), sub-protective (Southern Europe) and post-communist clusters. These closely follow the classification based on welfare state generosity, which is probably most used in studies concerning unemployed people. However, this study applies criteria based on a long, shared history (which means geographical proximity), religious communities and traditions, and language and forms five regional areas in Europe: Western, Northern, Southern, South-Eastern and Eastern [46]. For instance, these cluster characteristics can be seen in social norms to work. Stam et al. [44] used individual work ethic as a proxy for the social norm to work and showed higher scores in South-Eastern (with the influence of Islam) and Eastern (Communist heritage) European countries and lower scores in Northern Europe (Protestantism) On the other hand, work’s central role in identity and status can make unemployed people’s health and well-being more vulnerable in Protestant cultures than in different cultures [47]. European regions also differ in work values that are considered rather stable; however, there have been some variation according to economic fluctuations and corresponding unemployment rates. In fact, more variation may occur nationally between regions than between states, which makes comparisons between larger European regions difficult [48]. Work values can be distinguished as intrinsic or extrinsic [49,50,51] and are often positively correlated [51]. Intrinsic work values reflect autonomy (e.g., own initiative and work-family balance) whereas extrinsic values emphasize security (e.g., job security and high income). Extrinsic values seem to be highest in Southern and South-European Europe, and lowest in Northern Europe [49,51]. In intrinsic values, such a clear division has not been shown, however, they seem to be highest in Western Europe [52].

Welfare state generosity and labour market characteristics influence the well-being of unemployed people through several mechanisms. The level of unemployment protection paradoxically has two-way consequences; it has been shown that higher benefits are associated with better [53,54] but also with decreased well-being of the unemployed [9,55]. It has been argued that higher protection might prevent the unemployed from applying for work [43]. Unemployment benefits are one means of passive labour market policy (PLMP), whereas, for example, providing training is an active policy measure (ALMP); nevertheless, both have the potential to help unemployed people [43]. However, Nordic countries have succeeded in providing the best universally accessible training programs to enhance human capital and increase equal opportunities [56]. Nordic countries, representing “social democratic” welfare state regimes, have significantly declined their level of unemployment protection in recent decades. Currently, primarily western European “conservative” regimes, such as Luxembourg and Belgium, offer the highest levels of unemployment benefits [57]. In the “liberal” welfare regimes of the United Kingdom and Ireland, the level of unemployment benefits has traditionally been the lowest [56]. In addition to welfare state generosity, the well-being of the unemployed varies with economic fluctuations and the unemployment rate. Even though a high unemployment rate weakens well-being among both employed and unemployed people [43], it diminishes the well-being gap between them. This indicates that in times of high unemployment, the role of the social norm to work is smaller [44]. Moreover, life satisfaction and subjective well-being tend to be lower in urban areas compared to rural areas in Europe [58,59], which may also be a sign of a higher social norm to work, but also, more expensive living and life in general. A pathway between values and well-being can also be explained through social norms in general; when individuals live according to society’s prevailing norms, their well-being is better because their values face no confrontation. For example, people with pro-environmental behaviour, which stems from a green self-image value, are better off in countries with stronger shared green attitudes [60]. Furthermore, the link between values and behaviours differs across countries, and it is shown that in more conservative countries, the link is weaker [61].

Given that the selection process explains the poor health and well-being of unemployed people [7], it is suggested to be more substantial for women because they more often have physical and mental health problems that hinder their employment than men [21], who, in turn, more often report non-health-related reasons for unemployment [62]. Indeed, transitioning from school to work seems more complicated for young men than for women [63]. Moreover, unemployed women with strong traditional gender values, i.e., who think that work is dedicated to men rather than women, will be less likely to enter work compared to women with less traditional gender values. For example, in the Netherlands, this risk of not entering work was found to be greater in non-Western immigrant women than in native women [64]. In countries with more conservative gender values, the negative impact of unemployment is more significant for men than in countries with equalitarian gender values, where both genders are more equally hit by unemployment [9,65]. In general, unemployment tends to be more detrimental to men’s well-being than women’s [9,44]. This is explained by men’s traditional role as breadwinners and masculine identity, and on the other hand, the more prominent roles in domestic work and informal care activities among women [66]. However, the gap between the genders is closing in countries with more equal participation in market-based working life outside households [1].

This study explores the association between human values, self-rated health and subjective well-being among NEET young men and women across European regions. The research question is: How are human values associated with (a) self-rated health and (b) subjective well-being among NEETs across European regions and the two genders? We expect values and their association with self-rated health and subjective well-being to vary across European regions due to diverse cultural and societal factors. Moreover, we expect the association between values and self-rated health and subjective well-being to differ between genders because unemployment affects men and women differently. In addition, this study makes a valuable contribution to the literature by probing into the relation between contextual factors (such as social and cultural country characteristics) and how these factors influence the relation between values and health among NEET individuals. Moreover, for all relations under study, we use a gender perspective. To the best of our knowledge, no previous studies simultaneously take macro-level factors and gender issues into account when investigating the role of values for the health and well-being of NEETs.

## 2. Materials and Methods

### 2.1. Data and Participants

The study used data from rounds 5 to 9 of the European Social Survey (ESS) covering the years 2010–2018 correspondingly. Data collections with random probability sampling methods have been organised every two years since 2002 through face-to-face interviews among persons aged 15 and over. Datasets are freely available for researchers upon registration (https://ess-search.nsd.no/ (accessed on 1 September 2022)). The combined dataset included 241,222 participants from 34 European countries when Israel was excluded from this study. Israel was excluded because the used classification, the European Standard Classification of Cultural and Ethnic Groups [46], includes Israel in the group of North African, Middle Eastern and Central Asian nations. Of the participants, this study included 3842 young people aged 15–29 who were not in employment, education or training (NEET). We used only data from survey rounds 5 to 9, and survey round 10 in 2020 was excluded from this study because it was restricted due to the COVID-19 pandemic. Furthermore, it included only 233 NEETs and there were no Southern European participants. All the ESS rounds gathered information about human values, self-rated health and subjective well-being. Figure 1 and Figure 2 show their distributions by ESS round. The number of participants was most extensive in 2012 (n = 1102) and smallest in 2016 (n = 551).

### 2.2. Dependent Variables

Health was measured as Self-Rated Health (SRH), a 5-point Likert scale of a person’s subjective assessment of their current health status with the question: How is your health in general? Response options ranged from 1 (very bad) to 5 (very good). SRH is widely used in population studies since it reliably measures the current state of health and predicts mortality and morbidity [67,68]. We measured subjective well-being (SW) by combining two questions about satisfaction with life (All things considered, how satisfied are you with your life as a whole nowadays?) and happiness (Taking all things together, how happy would you say you are?). Both were answered with an 11-point scale ranging from 0 (extremely dissatisfied or extremely unhappy) to 10 (extremely satisfied or extremely happy), and we calculated the mean of the two questions. Life satisfaction represents a cognitive component, and happiness is an affective component of SW [69], whose internal consistency is assessed as Cronbach’s α = 0.80.

### 2.3. Independent Variables

The Human Values Scale contains 21 statements as value portraits [70]. Respondents were asked to rate how well the value portraits corresponded to them on a scale from 1 (not like me) to 6 (very much like me). The value portraits were organised under ten basic values, forming four higher-order values [35,71]. Means were calculated for each higher-order value. Table 1 shows how the basic values were structured in this study [71].

### 2.4. European Regions

The included 34 countries were grouped according to the guidance of the European Standard Classification of Cultural and Ethnic Groups (ESCEG), based on individuals’ specific socio-cultural and ethnic origins [46]. Five European areas were distinguished: Western (Austria, Belgium, Switzerland, Germany, France, United Kingdom, Ireland, Netherlands), Northern (Denmark, Finland, Iceland, Norway, Sweden), Southern (Cyprus, Spain, Greece, Italy, Portugal), South-Eastern (Albania, Bulgaria, Croatia, Montenegro, Serbia, Slovenia, Kosovo) and Eastern (Czechia, Estonia, Hungary, Lithuania, Latvia, Poland, Russian Federation, Slovakia, Ukraine) Europe.

### 2.5. Control Variables

We controlled for participation in five diverse activities which may replace the latent functions of work that the NEET young people are deprived of: time structure for the day, social contacts, personal status and identity, regular activity and participating in decision-making [13,15]. *Job-seeking* was whether a person was actively seeking a job and classified as 1 (No) or 2 (Yes). *Improving skills* included attending a course, lecture or conference during the last 12 months and classified 1 (No) and 2 (Yes). These two activities are not completely voluntary because, in several countries, active employment policies require the unemployed to demonstrate an active job search or to participate in various training courses. *Social activity* was measured with two items: the frequency of meeting friends or relatives was measured using a scale from 1 (never) to 7 (every day), and taking part in social activities compared to others of the same age used a scale from 1 (much less than most) to 5 (much more than most). The scales were normalised to a scale of 0–10 for comparability, and the mean was calculated. *Civic engagement* was calculated as a sum with dummies for the following—voting in the last national election and during the previous 12 months, contacting politicians or government officials, wearing or displaying campaign badges/stickers, signing petitions or boycotting certain products—with the scale ranging from 0 to 5. *Religious services* was the frequency of attending services apart from special occasions and assessed with a scale ranging from 1 (never) to 7 (every day). Furthermore, we controlled for age (mean), years spent in education (mean), disability classified as 1 (No) or 2 (Yes) and household’s income ranging from 1 (Very difficult on present income) to 4 (Living comfortably on present income).

### 2.6. Statistical Analyses

Participants’ characteristics and dependent and independent variables were described as means (sd) stratified by European region and gender (Table 2). To analyse the associations between higher-order values and SRH (Table 3) and SW (Table 4), linear regression models were run separately for each European region stratified by gender. Linear regressions were applied because they allowed us to examine the associations within regions better. For these models, R-squared estimates of variance are shown. Additionally, interactions between gender and the different values were calculated.

We calculated intraclass correlation coefficients (ICC) for SRH and SW to see how much of the variance is explained by the country. We could not reliably calculate ICCs for the European regions because only five to nine countries were in the region clusters. The ICC values across countries were 9% for SRH and 8% for SW, which was a reason to run a multilevel analysis. The multilevel models were run separately for men and women to estimate the betas for the association between values and SRH and SW (Table 5 and Table 6).

Moreover, activity variables were controlled in the models. First, interactions between the different values and regions were calculated for men and women separately, and then three-way interactions between gender, the different values and regions were calculated for the model including both genders (Model 5). All the models were adjusted for age, years in education and ESS round. The analyses, excluding the multilevel analyses, were conducted by applying design and population weights with complex sample procedures of SPSS version 27. As the raw weights cannot be used as such in the multilevel analyses, we could exclude them because the risk of having biased estimates is low with a large sample size [72].

## 3. Results

### 3.1. Participant Characteristics

The mean age of the NEET young people (n = 3842) was 23.6 (sd 3.3) years in the pooled data. Of the NEETs, 2100 were men and 1741 were women, and they had spent, on average, 12.7 (sd 3.1) years in full-time education. The data included participants from Western (n = 1167), Eastern (n = 885), Southern (n = 770), South-Eastern (n = 601) and Northern (n = 419) Europe. The South-Eastern Europeans had the shortest education, while Southern European women had the most extended. Participants reported having any kind of disability most in Northern Europe and least in Southern Europe. Living with household’s income was most comfortable in Northern Europe and most difficult in Eastern Europe. (Table 2). The SRH score was highest in South-Eastern European women and lowest in Northern European women. Subjective well-being, in turn, was best in Northern European men and poorest among Eastern European men. Overall, women had higher conservation value scores than men. South-Eastern women were the most conservative, and Northern European men were the least conservative. Openness to change was most pronounced among South-Eastern men and least among Eastern European women. South-Eastern men valued self-enhancement most, while Northern European women had the lowest scores. Self-transcendence was es-teemed most among Northern women and least among Eastern men. Moreover, job-seeking activity was highest in Northern European men and lowest in South-Eastern Europe. Western European women attended most of the skill-improving activities, and South-Eastern Europeans had the lowest skill-improving activity. In general, men were socially more active than women across all the regions, while women tended to participate more in religious services. Civic engagement was highest in Northern Europe and lowest in Eastern Europe.

The estimates from the linear regression analyses predicting SRH in NEET young men and women across European regions are presented in Table 3. In Western Europe, conservation among men and self-enhancement among women were positively and significantly associated with SRH. In Northern and Southern Europe, values were not significantly associated with SRH. Among South-Eastern European men, self-enhancement and self-transcendence were significantly associated with SRH. In Eastern Europe, self-enhancement and self-transcendence in women related positively to SRH. Overall, in all the European regions, self-enhancement predicted the best SRH, though mainly among women. Interactions showed that conservation in Western Europe and openness to change in Eastern Europe increased SRH more among men than among women. In turn, self-enhancement predicted better health among women than among men in Eastern Europe.

Furthermore, controlling activity showed that among Western men, social activity was significantly related to SRH. Active job-seeking was significantly associated with SRH among Northern European men, whereas social activity was positively associated and civic engagement was negatively associated with SRH in women. Furthermore, among Southern men, improving skills was positively associated with SRH, and social activity was positively associated with SRH in men and women in South-Eastern Europe. Moreover, improving skills, social activity and religious services among men and women were positively associated with SRH in Eastern Europe.

### 3.2. Subjective Well-Being and Values among NEETs across European Regions

Table 4 shows the regression analyses of the association between higher-order values and SW among NEET men and women stratified by European regions. In Western Europe, openness to change was positively associated with SW among men. Northern European women showed a positive association between conservation and a negative association of self-enhancement with SW. In Southern Europe, conservation related positively to SW in men. Conservation and openness to change had positive relations, and self-enhancement negatively affected SW among South-Eastern women. Finally, in Eastern Europe, self-transcendence related positively to SW among women. Interactions showed that self-transcendence predicted better SW in Western European and worse SW in South-Eastern European women compared to men (Table 4). Self-enhancement was associated more strongly with SW among Southern European women than among men. Moreover, among South-Eastern European women, conservation predicted better SW than among men.

Social activity and religious services were positively associated with SW among Western European men and women, while civic engagement had a negative association among men. In Northern Europe, improving skills was positively associated with SW among men and women, social activity among women only, and religious services had a negative association among men. Social activity related positively to SW among Southern European men and women. South-Eastern European men and women had positive associations between social activity and SW, and men had positive associations between improving skills and religious services and SW. Additionally, civic engagement was negatively associated with SW among South-Eastern European men. Women, in turn, showed negative associations between SW and improving skills and religious services. In Eastern Europe, social activity was positively associated with SW in men and women, as were religious services in men. Overall, it was found that improving skills, social activity and attending religious services were beneficial, and civic engagement was more or less harmful to SW across European regions.

### 3.3. Self-Rated Health of NEET Young Men and Women

The multilevel models in Table 5 present the estimates for the associations of higher-order values, activity behaviours and European regions with SRH stratified first by gender (Models 1–4). Model 5 shows results for both genders (Table 5). For men, conservation and openness to change had positive associations with SRH in Model 1, and the associations remained when all the variables were controlled (Model 4). All the activity behaviours, except improving skills, were associated either positively (job-seeking, social activity and religious services) or negatively (civic engagement) with SRH. Except for religious services, all stayed significant in the full model (Model 4). Model 3 shows that Eastern European men had significantly worse SRH than those in Southern Europe, but the association did not remain significant after controlling for all the variables.

Among women, self-enhancement was positively associated with SRH, even after adding all the variables in the model (Model 4). Social activity was positively and civic engagement negatively related to SRH, also in the full model. Moreover, South-Eastern European women had significantly better health than Eastern European women, and a significant connection remained after adding all the variables to the model. When testing two-way interactions value * region, it was shown that for Southern European women, self-transcendence predicted better SRH than for Eastern European women.

Model 5 shows that openness to change and self-enhancement were positively associated with SRH among men and women. Moreover, social activity predicted better SRH and civic engagement worse SRH among NEET men and women. No significant three-way interactions were found.

### 3.4. Subjective Well-Being of NEET Young Men and Women

Multilevel models predicting SW among NEET men and women are presented in Table 6. Among men, openness to change and self-transcendence were positively associated with SW, but the latter’s association did not remain significant after adjusting all the variables in Model 4. Improving skills, social activity, and religious services positively related to SW, and improving skills and social activity remained significant in the full model. Furthermore, Eastern European men had significantly lower SW than Western, Northern and South-Eastern men.

For women, openness to change and self-transcendence had positive associations, whereas self-enhancement had a significant negative association with SW (Table 6). A negative association of self-enhancement with SW remained significant after adding all the variables (Model 4). Social activity related positively to SW in a model where only activities were assessed and also in the full model. Moreover, Northern and South-Eastern European women had significantly higher SW than Eastern Europeans.

Model 5 shows that all the higher-order values were significant predictors of SW in both genders; all were positive except self-enhancement, which was negative. Additionally, improving skills, social activity and religious services were significantly associated with SW. Western European women had significantly better SW compared to Eastern women. The only significant interaction shows that for Western European men, openness to change was more important for SW than for women. Interactions showed that among Northern European men, openness to change predicted better SW than among Eastern European men.

## 4. Discussion

Our study explored the associations between human higher-order values and self-rated health (SRH) and subjective well-being (SW) among NEET young people across different regions in Europe. This is the first study to combine and explain health and well-being by values among young unemployed men and women. The obtained results increase the understanding of the factors affecting the health and well-being of NEET young people because values precede and guide actions and, thus, health behaviour, such as smoking, eating behaviour or seeking social support [40,45] and values can also influence how people feel, e.g., they can make you more or less trustful or, they can make you feel guilty or ashamed when you do not live according to prevailing norms. An individual’s opportunity to influence their health and health behaviour is, however, shaped and limited by several social, contextual and individual circumstances. This was seen in this study, as the relation between values and NEETs’ health and well-being differed significantly across European regions and genders; consequently, diverse values were associated with SRH and SW in different regions. Moreover, the values predicting SRH and SW in men and women varied.

Concerning the four higher-order values, women tended to be more conservative and less open to change across all regions than men. This finding might be explained by the actual or anticipated maternal role, which predicts a shift towards more conservative values at the expense of openness [73]. Additionally, self-enhancement was higher among men than among women in all the regions. Self-transcendence was, in turn, more prominent among women across Europe. The same pattern was seen among female students, who had more self-transcendence values, such as helpfulness and cooperation, compared to men in an environment (higher education) which enforces self-enhancement values, such as competitiveness and dominance; therefore, they had a lower sense of belonging and self-efficacy and were more likely gave up studies [74]. Furthermore, in Eastern Europe, conservation and self-enhancement were high, and openness to change and self-transcendence were low among both genders. Confirming the idea of conservation and self-enhancement as unhealthy values [40], we show the low SRH and SW among Eastern European NEETs [40]. This might reflect, first, the higher social norm to work in Eastern European countries [44]. Second, poor or non-existent unemployment protection and activation policies might play a role by decreasing SW in Eastern Europe.

Interestingly, SRH was the lowest in Northern European men and women, although they had the highest scores in openness and self-transcendence, i.e., healthy values. On the other hand, Northern European men and women had disabilities of any kind the most. This might result from a stronger health selection process to unemployment. However, SW was best in Northern Europe, which might in turn be partly due to the typically high trust in the region [75]. When exploring SRH across genders, we found that women reported lower SRH than men in all the regions. This might reflect the fact that women tend to be selected to unemployment because of poor health more than men [62]. Additionally, a higher level of education across European among women compared to men could support the health selection hypothesis because they fail in getting jobs despite being better educated. SW was higher among women across the regions, which is probably because women, in general, experience a lower social norm to work.

When analysing the association between values and SRH in European regions, we show that self-enhancement was positively related to SRH among Western European women, South-Eastern men and Eastern women. This is an interesting and rather contradictory finding because self-enhancement is considered an unhealthy value [40]. On the other hand, the unhealthiness has been shown in association with SW, not SRH. Our finding may stem from the distinct nature of the concepts of SW and SRH. As a reliable indicator of morbidity and mortality, SRH may reflect physical and mental capacity and potential participation in working life, which could benefit from a value stressing the active development of one’s own capabilities and status [37]. The positive association of self-transcendence, a healthy value, with SRH among South-Eastern men and Eastern European women followed previous results [40]. The positive associations of opposing values, self-enhancement and self-transcendence, in the two same groups indicates most likely the heterogeneity of NEETs. Interactions showed that in Western Europe, conservation predicted better SRH among men compared to women. Probably conservation leads to better health if men value ‘security’ within this dimension and being part of a family, assuming that social relations protect people’s health. In Eastern Europe, openness to change had a stronger association with SRH in men compared to women and self-enhancement related better to SRH among women than among men.

Multilevel models in men’s SRH show partly different results from previous research. Conservation and openness to change were positively related to SRH. It seems that both pathways—either restricting impulses and actions and valuing safety and traditions, or seeking experiences and excitement and valuing independence and creativity—may lead to good health. However, relations are presumably positive only when the prevailing values in society are not in conflict with personal values. Additionally, among women, contradictory as well, self-enhancement was positively associated with SRH. Nevertheless, young unemployed women seem to benefit from pursuing personal success and status, the ambitious features of the value. Our study also confirms the earlier results of the poorer health and well-being of NEETs compared to other young people [3]. Self-rated health (SRH) and subjective well-being (SW) were both significantly lower among the unemployed compared to students and employed people, and they were highest among students in the pooled sample.

Values related to SW were distinct from those related to SRH across regions and gender. Conservation was positively associated with SW among Northern and South-Eastern European women, which is also contradictory to earlier studies [9,40]. Perhaps this association could be explained by the different pathways, since conservation was lowest in Northern and highest in South-Eastern Europe. Northern European women might, instead of having a strong social norm to work, value safety and harmony, which enhance their SW, whereas Southern-European women might, in the culture of a strong social norm to work for men, be voluntarily oriented towards family. Moreover, openness to change related positively to SW among Western men and South-Eastern women. Openness to change reflects curiosity and novelty and might associate with future orientation, which was found to enhance SW [76]. Self-enhancement was negatively associated with SW in Northern and South-Eastern women. These two findings are in line with the expectations about the healthiness of these values [39]. The interactions confirmed the role of the social norm to work in men’s well-being because in South-Eastern Europe, conservation predicted SW more often among women than among men. It is interesting that in Southern Europe, self-enhancement was more significant for women’s SW compared to men’s; that might indicate that self-enhancement should remain passive rather than active among men, as suggested [40]. In the gender-stratified analyses, SW was associated positively with openness to change among men and negatively with self-enhancement among women. These findings are in line with the previous studies [39,40] and show that valuing personal achievements according to social standards, i.e., showing extrinsic motivation, is harmful to well-being. Therefore, we might partly confirm the argument that self-enhancement is a value which should remain passive rather than active, without behaviours possibly connected to it [40].

This study applied Jahoda’s deprivation model, which explains poor well-being among unemployed people by introducing activities that may replace work benefits as control variables [14]. We show that being socially active and meeting friends was positively related to both SRH and SW across Europe and among both genders. Introducing control variables in the models slightly attenuated the regression weights the values had with SRH and SW, and it is most likely that social activity moderated the effects. Higher social activity might be seen as a stronger motivation to improve skills [77], but we show mainly adverse effects of improving skills on SRH and SW, which was also shown in earlier research [77]. The negative effects of improving skills might be due to the fact that participation in training is mandatory and must be attended to be eligible for benefits. Job-seeking and improving skills were lowest in South-Eastern and Eastern Europe. This is interesting because job-seeking and improving skills could reflect self-enhancement, which was most prominent in those regions, and the willingness to enhance one’s status, thus reflecting extrinsic work values [49,51]. However, more probably these regions lack active and passive labour market policies, which might lead to passivity lengthening the duration of unemployment and increasing the feelings of hopelessness and frustration [26]. Frustration and missing opportunities to use and develop one’s skills, in turn, often turn into mistrust in society, which may be increased by blaming the system for not finding a job [28]. Lack of trust could contribute to attenuation of so-called healthy values, openness to change and self-transcendence. In addition, NEET young people need a lot of assistance from various stakeholders (e.g., mentors, employers, coaches) to empower them and find their capabilities and keys to healthy living [78]. A meta-analysis showed that interventions combining promotion of mental health (e.g., boosting self-confidence and coping with setbacks) and job-seeking skills could help unemployed people to boost their employment status [79].

### Limitations

While the multilevel models allowed us to explore the effects of socio-cultural factors across regions, we still lack the causal evidence and longitudinal data on the predictive impact of values on health and well-being due to the cross-sectional nature of the data. Further, we need to include the actual health behaviours that would clearly link values and health outcomes among unemployed people in the models. There is also a risk that the questions concerning values are answered in a socially acceptable way or according to prevailing values in society instead of personal views. The other risk is that more substantial health selection in some regions might seriously affect the results. For example, the fact that Northern European men and women reported the poorest health might be due to the exclusive and meritocratic labour market, with few low-skilled jobs available to apply for with health problems and limited ability to work. On the other hand, Eastern Europe’s deteriorated healthcare systems, increasing number of chronic diseases and lack of private healthcare services might lead to higher rates of ill-health among Eastern Europeans. Moreover, the earlier research linking values, health and well-being among NEET men and women is lacking; therefore, we had to assume the results concerning young people or unemployed people, in general, could be reflected in this study population. Additionally, we excluded data from ESS round 10 (2020) from this study because of the small number of participants and the COVID-19 pandemic. However, the results are still relevant today, as the values seem constant. Finally, we used voting in the previous elections as one item for Civic engagement. However, there were 86 persons under 18 years who probably had no right to vote.

## 5. Conclusions

Our findings contribute to a deeper understanding of individual, social and contextual determinants of health and well-being among young unemployed people (NEETs) across Europe. Because we found several relations between values and health and well-being that were contradictory to the earlier knowledge, it is likely that the social selection mechanism is present in many countries. This mechanism will select people with poor health for unemployment; therefore, other pathways (besides values) explain the relation between NEETs and poor health. The knowledge this study produced is highly relevant and interesting to communal actors in helping NEETs participate in the activities of nearby communities and national stakeholders in shaping policies that could increase their inclusion in the labour market and society. Moreover, we show that values matter and that the country’s context interacts with values; governments should be aware of their role in the relationship between individual values and health and establish policies that enable people to live according to their values. This study was an overview of the health and well-being of NEET young people across Europe. In the future, however, it would be useful to focus on each region’s economic, social, cultural and political conditions and meso-level factors, such as health services for the unemployed and individual-level characteristics. Future research should study the links between values and behaviour in more detail among NEETs and whether “unhealthy” values could be influenced. We also suggest that future research does longitudinal research (to exclude the social selection hypothesis) so that we can solely investigate the social causation hypothesis for NEET, value and health. Additionally, qualitative research could be necessary to study further the link between values and health and well-being among NEETs.

## Figures and Tables

**Figure 1 ijerph-20-04840-f001:**
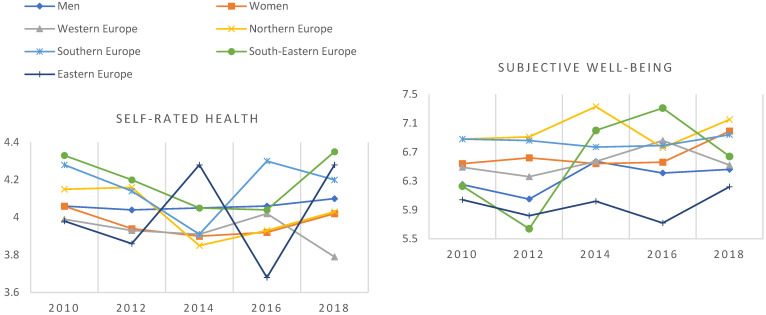
Self-rated health (scale 1–5) and subjective well-being (scale 0–10) by gender and European region among NEETs from 2010 to 2018 (ESS rounds 5–9).

**Figure 2 ijerph-20-04840-f002:**
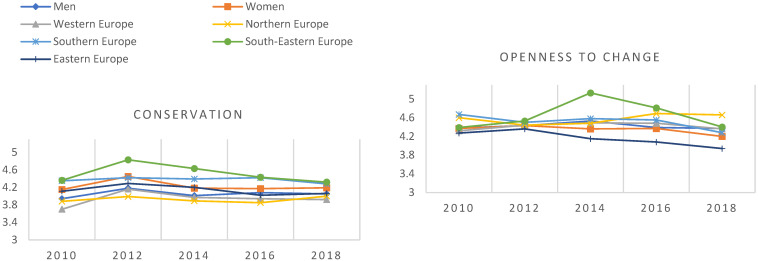
Higher-order values (scale 1–6) by gender and European region among NEETs from 2010 to 2020 (ESS rounds 5–10).

**Table 1 ijerph-20-04840-t001:** Values organised under ten basic values and four higher-order values.

Higher-Order Values	Basic Values	Example ^1^
**Conservation** α = 0.71 (Mean 4.18, sd 0.86)	Conformity	Restraint of actions, inclinations and impulses likely to upset or harm others and violate social expectations or norms (self-discipline, politeness, honouring parents and elders, obedience)
	Security	Safety, harmony and stability of society, relationships and self (family security, national security, social order, cleanliness, reciprocation of favours)
	Tradition	Respect for, commitment to and acceptance of the customs and ideas that traditional culture or religion provide (devout, respect for tradition, humble, moderate)
**Openness to change** α = 0.75 (Mean 4.44, sd 0.83)	Hedonism	Pleasure or sensuous gratification for oneself (pleasure, enjoying life, self-indulgent)
	Self-direction	Independent thought and action—choosing, creating, exploring (creativity, freedom, independence, choosing own goals, curiosity)
	Stimulation	Excitement, novelty and challenge in life (daring, a varied life, an exciting life)
**Self-enhancement** α = 0.70 (Mean 3.92, sd 0.99)	Achievement	Personal success through demonstrating competence according to social standards (ambitious, successful, capable, influential)
	Power	Social status and prestige, control or dominance over people and resources (authority, social power, wealth, preserving my public image)
**Self-transcendence** α = 0.76 (Mean 4.78, sd 0.79)	Benevolence	Preservation and enhancement of the welfare of people with whom one is in frequent personal contact (helpful, honest, forgiving, loyal, responsible)
	Universalism	Understanding, appreciation, tolerance, and protection of the welfare of all people and nature (equality, social justice, wisdom, broadmindedness, protecting the environment, unity with nature, a world of beauty)

^1^ Examples from Schwartz and Boehnke [71].

**Table 2 ijerph-20-04840-t002:** Descriptive statistics (Mean, SE) stratified by gender and European region.

	Western Europe		Northern Europe		Southern Europe		South-Eastern Europe		Eastern Europe	
	Men	Women	Men	Women	Men	Women	Men	Women	Men	Women
	n = 658	n = 509	n = 248	n = 171	n = 378	n = 391	n = 329	n = 272	n = 487	n = 398
	M (SE)	M (SE)	M (SE)	M (SE)	M (SE)	M (SE)	M (SE)	M (SE)	M (SE)	M (SE)
**Background characteristics**										
Age, years	23.46 (0.13)	23.59 (0.16)	23.05 (0.22)	23.61 (0.26)	23.93 (0.16)	24.09 (0.20)	23.62 (0.19)	24.02 (0.22)	23.53 (0.15)	23.65 (0.17)
Education, years	12.96 (0.12)	13.19 (0.14)	12.72 (0.17)	12.73 (0.27)	12.80 (0.20)	13.51 (0.22)	11.59 (0.17)	12.35 (0.21)	12.35 (0.12)	12.73 (0.13)
Disability (1 = No, 2 = Yes)	1.16 (0.01)	1.17 (0.02)	1.23 (0.03)	1.36 (0.04)	1.02 (0.01)	1.06 (0.01)	1.12 (0.02)	1.09 (0.02)	1.13 (0.02)	1.16 (0.02)
Household’s income (scale 1–4)	2.54 (0.04)	2.63 (0.04)	2.98 (0.06)	2.78 (0.07)	2.40 (0.05)	2.32 (0.05)	2.38 (0.05)	2.42 (0.06)	2.16 (0.04)	2.31 (0.04)
**Dependent variables**										
Self-rated health (scale 1–5)	4.13 (0.03)	4.03 (0.04)	4.06 (0.05)	3.93 (0.07)	4.28 (0.03)	4.09 (0.05)	4.27 (0.05)	4.31 (0.05)	4.11 (0.04)	4.01 (0.04)
Subjective well-being (scale 0–10)	6.50 (0.08)	6.63 (0.09)	7.19 (0.11)	6.99 (0.16)	6.71 (0.12)	7.03 (0.10)	6.25 (0.15)	6.84 (0.14)	5.80 (0.10)	6.31 (0.11)
**Higher-order values (scales 1** **–** **6)**										
Conservation	3.95 (0.03)	4.14 (0.04)	3.81 (0.06)	4.10 (0.06)	4.28 (0.04)	4.49 (0.04)	4.47 (0.05)	4.66 (0.05)	4.01 (0.04)	4.25 (0.04)
Openness to change	4.42 (0.03)	4.42 (0.04)	4.54 (0.05)	4.59 (0.06)	4.52 (0.04)	4.47 (0.05)	4.65 (0.05)	4.39 (0.06)	4.28 (0.04)	4.26 (0.05)
Self-enhancement	3.81 (0.04)	3.66 (0.04)	3.56 (0.06)	3.43 (0.08)	3.94 (0.05)	3.89 (0.06)	4.35 (0.05)	4.04 (0.06)	4.06 (0.04)	4.09 (0.05)
Self-transcendence	4.68 (0.03)	4.90 (0.03)	4.91 (0.04)	5.18 (0.05)	4.90 (0.04)	5.10 (0.04)	4.83 (0.04)	4.97 (0.05)	4.33 (0.04)	4.60 (0.04)
**Activity**										
Active job-seeking (n, %)										
Yes	498 (76.4)	368 (72.9)	199 (80.2)	131 (78.4)	301 (79.6)	309 (79.8)	224 (68.5)	177 (65.8)	359 (73.9)	268 (69.8)
No	154 (23.6)	137 (27.1)	49 (19.8)	36 (21.6)	77 (20.4)	78 (20.2)	103 (31.5)	92 (34.3)	127 (26.1)	120 (30.2)
Improving skills during the 12 last months (n, %)										
Yes	435 (66.1)	355 (70.2)	99 (40.1)	73 (42.7)	82 (21.8)	91 (23.4)	45 (13.8)	52 (19.1)	84 (17.4)	99 (25.1)
No	223 (33.9)	151 (29.8)	148 (59.9)	98 (57.3)	295 (78.2)	298 (76.6)	282 (86.2)	220 (80.9)	399 (82.6)	295 (74.9)
Social activity (scale 1–7)	5.88 (0.08)	5.57 (0.09)	6.26 (0.11)	5.89 (0.15)	6.42 (0.09)	5.89 (0.13)	6.24 (0.12)	5.42 (0.14)	5.72 (0.10)	5.52 (0.10)
Civic engagement (scale 0–5)	0.91 (0.04)	0.87 (0.05)	1.37 (0.07)	1.61 (0.09)	0.91 (0.06)	1.00 (0.06)	0.85 (0.05)	0.80 (0.05)	0.59 (0.04)	0.70 (0.05)
Religious services (scale 1–7)	2.06 (0.06)	2.12 (0.06)	1.85 (0.07)	1.99 (0.10)	2.15 (0.08)	2.46 (0.10)	2.65 (0.07)	2.60 (0.08)	2.17 (0.06)	2.65 (0.08)

**Table 3 ijerph-20-04840-t003:** Unstandardised betas (standard errors) for the association between SRH and values in NEET young men and women stratified by European region ^a^.

	Western Europe		Northern Europe		Southern Europe		South-Eastern Europe		Eastern Europe	
	Men	Women	Men	Women	Men	Women	Men	Women	Men	Women
	B (Se)	B (Se)	B (Se)	B (Se)	B (Se)	B (Se)	B (Se)	B (Se)	B (Se)	B (Se)
**Intercept**	3.33 (0.94) ***	5.14 (1.03) ***	1.91 (1.00) ***	0.17 (1.06)	4.41 (0.65) ***	5.88 (1.08) ***	4.01 (0.77) ***	2.32 (0.85) ***	6.46 (0.80) ***	3.46 (0.75) ***
**Higher-order values**										
Conservation	0.11 (0.05) *	−0.09 (0.06)	−0.09 (0.07)	0.10 (0.10)	0.05 (0.06)	0.09 (0.10)	−0.02 (0.05)	−0.07 (0.05)	0.07 (0.04)	0.01 (0.06)
Openness to change	0.07 (0.05)	0.09 (0.05)	0.06 (0.09)	−0.06 (0.11)	0.04 (0.07)	0.03 (0.08)	0.07 (0.04)	−0.01 (0.03)	0.05 (0.04)	−0.08 (0.05)
Self-enhancement	0.04 (0.04)	0.09 (0.05) *	−0.05 (0.06)	0.07 (0.08)	0.00 (0.05)	0.06 (0.06)	0.07 (0.03) *	0.02 (0.03)	0.04 (0.03)	0.17 (0.05) ***
Self-transcendence	0.02 (0.05)	0.05 (0.07)	0.15 (0.14)	0.12 (0.12)	0.05 (0.07)	−0.09 (0.10)	0.16 (0.05) **	0.05 (0.04)	−0.03 (0.05)	0.14 (0.07) *
**Activity**										
Job-seeking	0.16 (0.09)	−0.15 (0.08)	0.55 (0.14) ***	−0.18 (0.18)	−0.00 (0.07)	−0.23 (0.12)	−0.04 (0.06)	−0.03 (0.05)	−0.03 (0.06)	0.06 (0.07)
Improving skills	0.12 (0.07)	0.08 (0.07)	−0.06 (0.10)	0.28 (0.15)	−0.25 (0.07) **	−0.22 (0.13)	0.10 (0.10)	−0.09 (0.08)	−0.23 (0.10) *	−0.22 (0.07) **
Social activity	0.05 (0.02) **	0.03 (0.02)	0.07 (0.04)	0.15 (0.05) **	0.08 (0.02)	0.04 (0.04)	0.06 (0.01) ***	0.05 (0.01) ***	0.06 (0.01) ***	0.07 (0.02) **
Civic engagement	−0.02 (0.03)	−0.07 (0.04)	−0.07 (0.05)	−0.15 (0.06) *	−0.05 (0.03)	0.03 (0.05)	−0.06 (0.04)	0.06 (0.03)	−0.04 (0.04)	−0.06 (0.05)
Religious services	0.05 (0.03)	−0.03 (0.03)	−0.01 (0.05)	0.07 (0.06)	0.03 (0.02)	−0.07 (0.04)	0.00 (0.02)	0.02 (0.02)	0.08 (0.01) ***	0.07 (0.02) ***
R^2^	0.14	0.09	0.19	0.28	0.13	0.12	0.20	0.23	0.22	0.20
**Interactions ^b^**										
Women * conservation		−0.19 (0.07) **								
Women * openness to change										−0.18 (0.05) **
Women * self-enhancement										0.15 (0.05) **

^a^ Models controlled for age, years in education, countries and ESS survey waves. * *p* < 0.05, ** *p* < 0.01, *** *p* < 0.001. ^b^ Interactions gender * value. Only significant interactions are shown.

**Table 4 ijerph-20-04840-t004:** Unstandardised betas (standard errors) for the association between SW and values in NEET men and women stratified by European region ^a^.

	Western Europe		Northern Europe		Southern Europe		South-Eastern Europe		Eastern Europe	
	Men	Women	Men	Women	Men	Women	Men	Women	Men	Women
	B (Se)	B (Se)	B (Se)	B (Se)	B (Se)	B (Se)	B (Se)	B (Se)	B (Se)	B (Se)
**Intercept**	3.42 (1.87) ***	5.56 (2.19) ***	8.95 (2.20) ***	3.85 (2.31)	5.51 (2.26) ***	4.25 (2.12) *	1.71 (1.98) ***	−6.53 (1.61) ***	10.94 (2.02) ***	4.18 (2.30) *
**Higher-order values**										
Conservation	0.05 (0.09)	−0.04 (0.13)	0.03 (0.16)	0.55 (0.24) *	0.34 (0.16) *	0.10 (0.21)	−0.20 (0.12)	0.42 (0.11) ***	0.10 (0.14)	−0.04 (0.15)
Openness to change	0.43 (0.16) **	−0.11 (0.11)	0.20 (0.19)	0.29 (0.25)	0.13 (0.21)	0.13 (0.15)	0.13 (0.12)	0.29 (0.06) ***	−0.19 (0.14)	0.00 (0.12)
Self-enhancement	−0.11 (0.10)	−0.04 (0.10)	−0.15 (0.14)	−0.39 (0.18) *	−0.32 (0.17)	0.11 (0.13)	−0.07 (0.08)	−0.15 (0.05) **	0.12 (0.11)	0.00 (0.13)
Self-transcendence	−0.07 (0.15)	−0.01 (0.14)	0.28 (0.23)	−0.06 (0.31)	0.22 (0.22)	0.19 (0.19)	0.17 (0.10)	−0.03 (0.10)	0.21 (0.14)	0.60 (0.18) **
**Activity**										
Job-seeking	0.10 (0.20)	−0.07 (0.17)	0.28 (0.31)	0.06 (0.37)	−0.21 (0.23)	−0.22 (0.24)	−0.04 (0.14)	−0.15 (0.10)	0.23 (0.15)	−0.00 (0.19)
Improving skills	0.18 (0.16)	0.00 (0.15)	0.72 (0.23) **	0.74 (0.34) *	0.18 (0.23)	0.09 (0.24)	0.93 (0.23) ***	−0.45 (0.19) *	−0.02 (0.26)	−0.13 (0.17)
Social activity	0.12 (0.05) *	0.27 (0.05) ***	0.02 (0.08)	0.33 (0.12) **	0.18 (0.08) *	0.16 (0.06) **	0.27 (0.02) ***	0.25 (0.03) ***	0.15 (0.03) ***	0.19 (0.06) **
Civic engagement	−0.19 (0.08) *	0.11 (0.09)	−0.17 (0.11)	−0.08 (0.14)	0.09 (0.12)	−0.12 (0.11)	−0.42 (0.12) ***	0.11 (0.10)	−0.11 (0.09)	−0.11 (0.13)
Religious services	0.20 (0.06) **	0.14 (0.06) *	−0.28 (0.11) *	0.11 (0.11)	−0.00 (0.08)	0.03 (0.07)	0.11 (0.05) *	−0.12 (0.03) **	0.23 (0.04) ***	0.06 (0.03)
R^2^	0.18	0.14	0.17	0.23	0.11	0.11	0.27	0.29	0.16	0.13
**Interactions ^b^**										
Women * conservation								0.67 (0.13) ***		
Women * openness to change										
Women * self-enhancement						0.51 (0.20) *				
Women * self-transcendence		0.46 (0.18) *						−0.44 (0.13) **		

^a^ Models controlled for age, years in education, countries and ESS survey waves. * *p* < 0.05, ** *p* < 0.01, *** *p* < 0.001. ^b^ Interactions gender * value. Only significant interactions are shown.

**Table 5 ijerph-20-04840-t005:** Multilevel models ^a^ for the associations between SRH and values in NEET men (n = 2100). and women (n = 1742).

	Men				Women				Total ^b^
	Model 1	Model 2	Model 3	Model 4	Model 1	Model 2	Model 3	Model 4	Model 5
	B (Se)	B (Se)	B (Se)	B (Se)	B (Se)	B (Se)	B (Se)	B (Se)	B (Se)
**Intercept**	3.60 (0.25) ***	3.87 (0.23) ***	4.50 (0.37) ***	3.28 (0.41) ***	3.43 (0.29) ***	3.72 (0.27) ***	4.54 (0.45) ***	3.77 (0.48) ***	3.61 (0.37) ***
**Higher-order values**									
Conservation	0.06 (0.03) *			0.06 (0.03) *	0.00 (0.03)			0.01 (0.03)	0.03 (0.02)
Openness to change	0.11 (0.03) ***			0.09 (0.03) **	0.00 (0.03)			−0.00 (0.03)	0.04 (0.02) *
Self-enhancement	0.04 (0.02)			0.03 (0.02)	0.07 (0.03) **			0.06 (0.04) *	0.05 (0.02) **
Self-transcendence	0.00 (0.03)			0.01 (0.03)	0.05 (0.04)			0.06 (0.04)	0.03 (0.02)
**Activity**									
Job-seeking		0.12 (0.04) **		0.11 (0.04) **		−0.07 (0.04)		−0.08 (0.04)	0.03 (0.03)
Improving skills		0.00 (0.04)		0.00 (0.04)		−0.04 (0.05)		−0.05 (0.05)	−0.01 (0.03)
Social activity		0.06 (0.01) ***		0.05 (0.01) ***		0.05 (0.01) ***		0.05 (0.01) ***	0.06 (0.01) ***
Civic engagement		−0.05 (0.02) **		−0.04 (0.02) *		−0.06 (0.02) **		−0.06 (0.02) **	−0.05 (0.01) ***
Religious services		0.03 (0.01) **		0.02 (0.01)		0.01 (0.01)		−0.00 (0.01)	0.01 (0.01)
**European regions**									
Western Europe			0.09 (0.13)	0.09 (0.13)			0.06 (0.16)	0.01 (0.16)	0.05 (0.13)
Northern Europe			0.02 (0.11)	0.04 (0.11)			−0.03 (0.14)	−0.04 (0.13)	−0.01 (0.11)
Southern Europe			−0.30 (0.13) *	−0.22 (0.13)			−0.32 (0.15)	−0.30 (0.14)	−0.27 (0.13) *
South-Eastern Europe			−0.19 (0.12)	−0.10 (0.12)			−0.35 (0.15) *	−0.33 (0.14) *	−0.22 (0.12)
Eastern Europe (ref.)									
**Interactions ^c^**									
Self-transcendence * Southern Europe								0.21 (0.10) *	
**Interactions ^d^**									

^a^ Models controlled for age, years in education and ESS survey waves. ^b^ Model 5 includes both men and women. ^c^ Interactions value * region for Models 4, only significant interaction are shown. ^d^ Interactions gender * value * region for Model 5, only significant interaction are shown. * *p* < 0.05, ** *p* < 0.01, *** *p* < 0.001.

**Table 6 ijerph-20-04840-t006:** Multilevel models ^a^ for the associations between SW and values in NEET men (n = 2100) and women (n = 1742).

	Men				Women				Total
	Model 1	Model 2	Model 3	Model 4	Model 1	Model 2	Model 3	Model 4	Model 5 ^b^
	**B (Se)**	**B (Se)**	**B (Se)**	**B (Se)**	**B (Se)**	**B (Se)**	**B (Se)**	**B (Se)**	**B (Se)**
**Intercept**	5.72 (0.68) ***	5.80 (0.62) ***	10.24 (0.95) ***	6.68 (1.06) ***	3.96 (0.72) ***	4.43 (0.66) ***	7.39 (0.93) ***	4.39 (1.05) ***	5.82 (0.87) ***
**Higher-order values**									
Conservation	0.02 (0.07)			0.07 (0.07)	0.08 (0.08)			0.14 (0.08)	0.12 (0.05) *
Openness to change	0.24 (0.07) **			0.14 (0.07) *	0.19 (0.08) *			0.14 (0.07)	0.12 (0.05) *
Self-enhancement	−0.05 (0.06)			−0.07 (0.06)	−0.15 (0.06) *			−0.15 (0.06) *	−0.12 (0.04) **
Self-transcendence	0.15 (0.08) *			0.11 (0.08)	0.19 (0.09) *			0.16 (0.09)	0.16 (0.06) **
**Activity**									
Job-seeking		0.05 (0.10)		0.02 (0.10)		−0.06 (0.11)		−0.11 (0.11)	−0.06 (0.08)
Improving skills		0.29 (0.11) **		0.28 (0.11) *		0.10 (0.12)		0.08 (0.12)	0.18 (0.08) *
Social activity		0.20 (0.02) ***		0.19 (0.02) ***		0.20 (0.03) ***		0.20 (0.03) ***	0.19 (0.02) ***
Civic engagement		−0.06 (0.05)		−0.09 (0.05)		0.01 (0.05)		−0.02 (0.05)	−0.07 (0.04)
Religious services		0.07 (0.03) *		0.06 (0.03)		0.03 (0.04)		0.03 (0.04)	0.06 (0.03) *
**European regions**									
Western Europe			−1.29 (0.32) ***	−1.11 (0.32) **			−0.66 (0.30) *	−0.41 (0.30)	−0.80 (0.29) *
Northern Europe			−0.72 (0.28) *	−0.58 (0.28) *			−0.46 (0.24)	−0.34 (0.24)	−0.50 (0.25)
Southern Europe			−0.61 (0.32)	−0.39 (0.31)			−0.45 (0.27)	−0.29 (0.26)	−0.35 (0.28)
South-Eastern Europe			−0.65 (0.30) *	−0.43 (0.29)			−0.58 (0.27) *	−0.44 (0.26)	−0.42 (0.26)
Eastern Europe (ref.)									
**Interactions ^c^**									
Openness to change * Northern Europe				−0.56 (0.19) **					
**Interactions ^d^**									
Men * openness to change * Northern Europe									−0.49 (0.19) **

^a^ Models controlled for age, years in education and ESS survey waves. ^b^ Model 5 includes both men and women. ^c^ Interactions value * region for Models 4, only significant interaction are shown. ^d^ Interactions gender * value * region for Model 5, only significant interaction are shown. * *p* < 0.05, ** *p* < 0.01, *** *p* < 0.001.

## Data Availability

Datasets are freely available for researchers upon registration (https://ess-search.nsd.no/ (accessed on 1 September 2022)).

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
