# Peer review of "Values, Health and Well-Being of Young Europeans Not in Employment, Education or Training (NEET)"

_ijerph, 2023, doi:10.3390/ijerph20064840_

Round 1
Reviewer 1 Report
Overall, we believe the objective of the paper is correctly posed and offers an interesting starting point to explore relationship between human values, self-rated health, and subjective wellbeing in young unemployed in Europe considering gender perspective and values of different socio-cultural regions. However, more explanation of the literature review is needed to link RQs with the discussion section.
The main point is that the article is an exploratory study which presents two research questions, but the discussion section is more focused in expected results that are not totally consistent with the evidence and arguments presented in the literature review.
Looking at Methods section, the manuscript does not offer enough information about the instruments used to measure the dependent variables, it would be advisable to give an example of item. Regarding some variables that impact unemployment, it would be advisable to get some information about other variables that could impact or moderate the results in the well-being of unemployed, as family obligations (at least, having or not children) or financial deprivation, and disability as an important background characteristic. Finally, we would like to mention an observation about the control variable "Civic engagement", as it is measured by 4 items, one of them “voting in the last national election and during the previous 12 months”. One part of the sample of the study, 15-17, are not allowed to vote until 18, in many European countries.
The literature review in this paper mentions many different theories, but It would be necessary for the total understanding of the paper to go in depth at least with the assertion that values precede and guide actions and, thus, health behavior; social selection of unemployment and social causation process theories. On the other hand, we consider important going in depth in cultural and societal factors in each regional country to provide explanation about the results, and to consider meaning of work /work values in the literature review.
Finally, we want to mention some specific comments to the authors:
Line 45 It is unclear that it could be asserted directly that disability of any kind is directly a health problem.
Line 105 It lacks some reference.
Line 212-213 It lacks some reference.
Line 216-218 The meaning of the sentence is not clear.
Line 496 It lacks some reference
Finally, and despite the fact the conclusions have correctly set out the limitations, it is suggested to include clearly future lines of research and more specific practice implications.
Lastly, overall, I would like to stress that the authors are working in a fascinating area and I encourage them to continue their important stream of research.
Reviewer 2 Report
The subject matter is interesting.
The following comments on the manuscript are made below:
Perhaps it would have been more appropriate to carry out a study on a European region, analysing in greater depth its social, economic, social, cultural and political circumstances, ...
Jahoda's latent depression model should be briefly described.
The reason for the choice of the period studied should be justified.
The following idea needs to be better explained: Pooled European Social Survey data from 2010‒2018 were used, and multilevel analyses with region dummies stratified by gender and linear regressions stratified by European socio-cultural region and gender with interactions were performed.
The objective of the work and the gap to be covered in the literature need to be clearly established. In other words, explain the purpose of carrying out this type of work.
The added value of the work needs to be clearly stated.
Pointing out what the theoretical contribution to the literature is foundational, mental, as well as the practical implications.
For a better understanding of the manuscript, a better contextualisation of the studied areas is necessary. In addition, it would be interesting to study whether the results obtained are related to social, economic and political circumstances.
It is recommended that the conclusions be improved. They are brief and could be expanded by further highlighting the results of the work.
The practical implications and future lines of research are unclear.
Some recommended bibliography:
Wanberg, C.R.,Ali,A.A, Csillag, B. (2020).
Job Seeking: The Process and Experience of Looking for a Job. Annu. Rev. Organ. Psychol. Organ. Behav. 2020. 7:315–37.
Felaco, C., Parola, A. (2022). Subjective Well-Being and Future Orientation of NEETs: Evidence from the Italian Sample of the European Social Survey. Social Sciences 11: 482. https://doi.org/ 10.3390/socsci11100482
